# Engineered migrasomes provide a robust and thermally stable vaccination platform

Dongju Wang[1†], Haifang Wang[2,3†], Wei Wan[2], Zihui Zhu[4], Takami Sho[1], Yi Zheng[1], Xing Zhang[5], Longyu Dou[6], Qiang Ding[4], Li Yu[1]*, Zhihua Liu[2]*

[1]The State Key Laboratory of Membrane Biology, Tsinghua-Peking Joint Center for Life Sciences, Beijing Frontier Research Center for Biological Structure, School of Life Sciences, Tsinghua University, Beijing, China; [2]The State Key Laboratory of Membrane Biology, Tsinghua-Peking Joint Center for Life Sciences, Institute of Immunology, School of Basic Medical Sciences, Tsinghua University, Beijing, China; [3]Clinical Stem Cell Research Center, Peking University Third Hospital, Beijing, China; [4]School of Basic Medical Sciences, Tsinghua University, Beijing, China; [5]Ministry of Education Key Laboratory of Protein Sciences, Tsinghua-Peking Joint Center for Life Sciences, Beijing Advanced Innovation Center for Structural Biology, Beijing Frontier Research Center for Biological Structures, School of Life Science, Tsinghua University, Beijing, China; [6]Migrasome Therapeutics, Beijing, China

*For correspondence:
liyulab@mail.tsinghua.edu.cn
(LY);
zhihualiu@mail.tsinghua.edu.
cn (ZL)

[†]These authors contributed
equally to this work.

Competing interest: See page
19

Reviewing Editor: Urszula
Krzych, Walter Reed Army
Institute of Research, United
States

## eLife Assessment

This study, from the group that pioneered migrasome, describes a novel vaccine platform of engineered migrasomes that behave like natural migrasomes. Importantly, this platform has the potential to overcome obstacles associated with cold chain issues for vaccines such as mRNA. In the revised version, the authors have addressed previous concerns, and the results from additional experiments provide **compelling** evidence that features methods, data, and analyses more rigorous than the current state-of-the-art. Although the findings are **important** with practical implications for the vaccine technology, results from additional experiments would make this an outstanding study.

**Abstract** The growing ability of pathogens and tumor cells to evade immune surveillance underscores the urgent need for new vaccine platforms that harness diverse biological mechanisms. Logistical constraints associated with cold-chain transport further limit vaccine accessibility, particularly in resource-limited settings. Migrasomes—specialized organelles produced during cell migration—are inherently stable and enriched with immune-modulating molecules. To overcome the low yield of natural migrasomes, we engineered migrasome-like vesicles (eMigrasomes) using hypotonic shock combined with cytoskeletal disruption to promote vesicle formation. eMigrasome biogenesis depends on core migrasome machinery and recapitulates the biophysical and molecular features of native migrasomes while achieving higher production efficiency. In murine models, eMigrasomes loaded with a model antigen elicited potent antibody responses and retained structural integrity and immunogenicity at room temperature. Moreover, eMigrasomes displaying the SARS-CoV-2 Spike protein induced strong humoral responses and conferred protection against viral challenge in mice. These results establish eMigrasomes as an innovative, thermally stable, and broadly applicable vaccine platform derived from migrasome biology.

## Introduction

To face the imminent challenges of emerging infectious diseases, and to realize the not-yet-fulfilled promise of cancer vaccines, new types of vaccination platforms based on different basic biological principles are urgently needed. The huge success of mRNA vaccines in the fight against the Covid-19 pandemic further supports the importance of developing vaccine platforms based on different underlying biological mechanisms (*Park et al., 2021*; *Polack et al., 2020*; *Baden et al., 2021*; *Gui et al., 2023*). However, the application of mRNA-based vaccines, or other forms of traditional vaccine, has been hampered by the requirement for complicated cold-chain transportation, in which the materials are constantly maintained at low temperature (*Crommelin et al., 2021*; *Grau et al., 2021*). For most parts of the world, especially the Global South, it is vitally important to develop vaccines that do not require cold-chain transportation.

Migrasomes are recently discovered organelles of migrating cells (*Ma et al., 2015*). During migrasome formation, long membrane tethers named retraction fibers are pulled out at the trailing edge of migrating cells by the force generated by the movement of the cells. Migrasomes are formed on retraction fibers by a complicated process involving multiple components (*Wu et al., 2017*; *Huang et al., 2019*; *Ding et al., 2023*). In the final expansion step, the growth of migrasomes is driven by assembly of nanometer-scaled tetraspanin-enriched microdomains (TEMs) into micrometer-scaled tetraspanin-enriched macrodomains (TEMAs) (*Huang et al., 2019*; *Dharan et al., 2023*). Thus, migrasomes are highly enriched with components of TEMs—such as tetraspanins, integrins, and cholesterol—and formation of migrasomes is dependent on the presence of these molecules. For example, overexpression of Tspan4 can promote migrasome formation (*Huang et al., 2019*), while removing cholesterol can block migrasome formation. Theoretical modeling, in vitro reconstitution of migrasome formation, and membrane stiffness measurement by atomic force microscopy have revealed mechanistic insights into migrasome formation. These approaches showed that tetraspanin- and cholesterol-enriched macrodomains have highly elevated membrane stiffness, which is the key factor to drive the bulging of retraction fibers into migrasomes; as a result, migrasomes are highly rigid (*Huang et al., 2019*). More recently, it was shown that assembly of TEMs can repair damaged membranes by restricting the spread of membrane rupture. In liposomes containing Tspan4 and cholesterol, detergent-induced membrane damage can be rapidly repaired, and Tspan4-embedded liposomes are highly resistant to damage (*Huang et al., 2022*).

One defining feature of TEMs is the enrichment of immune-modulating molecules such as members of the immunoglobulin superfamily (IgSF) (*Levy and Shoham, 2005*; *Charrin et al., 2003*; *Clark et al., 2001*). This group of molecules contains many key signaling molecular complexes that regulate the immune response, including antigen receptors, antigen-presenting molecules, co-receptors, antigen receptor accessory molecules, and co-stimulatory or inhibitory molecules (*Lefranc and Lefranc, 2001*). Since migrasomes are largely composed of TEMs, members of the IgSF are also enriched in migrasomes. This property, in addition to the high stability of migrasomes, prompted us to explore the possibility of developing a migrasome-based vaccine.

One key obstacle to developing a migrasome-based delivery system and migrasome-based vaccines is the very low yield of migrasomes. To generate migrasomes, cells must be grown at very low density to allow them to migrate, and a migrating cell can only generate a relatively low number of migrasomes in a process that takes hours to finish (*Ma et al., 2015*). In this study, by using the biophysical insights we gained from investigating migrasome formation, we successfully overcame this key obstacle. We developed a method that mimics the key biophysical features of migrasome formation while vastly improving the yield of migrasome-like vesicles. We named these vesicles as engineered migrasomes (eMigrasomes). Using eMigrasomes loaded with the model antigen ovalbumin (OVA), we successfully demonstrate that eMigrasomes are a highly effective, room temperature (RT)-stable vaccine platform which generate a strong immunoglobulin G (IgG) response. Finally, we demonstrate that eMigrasomes loaded with SARS-CoV-2 Spike protein (S protein) can generate a strong protective humoral response against SARS-CoV-2. In summary, our study demonstrates that the eMigrasome-based vaccine platform is stable at RT, highly effective, and uses an unconventional immunological mechanism, which could be useful in certain scenarios where conventional vaccine platforms are less successful.

## Results

By serendipity, we found that hypotonic shock induces rapid formation of migrasome-like structures on retraction fibers in Tspan4-GFP-expressing cells (*Figure 1A*). Similar phenomena had also been

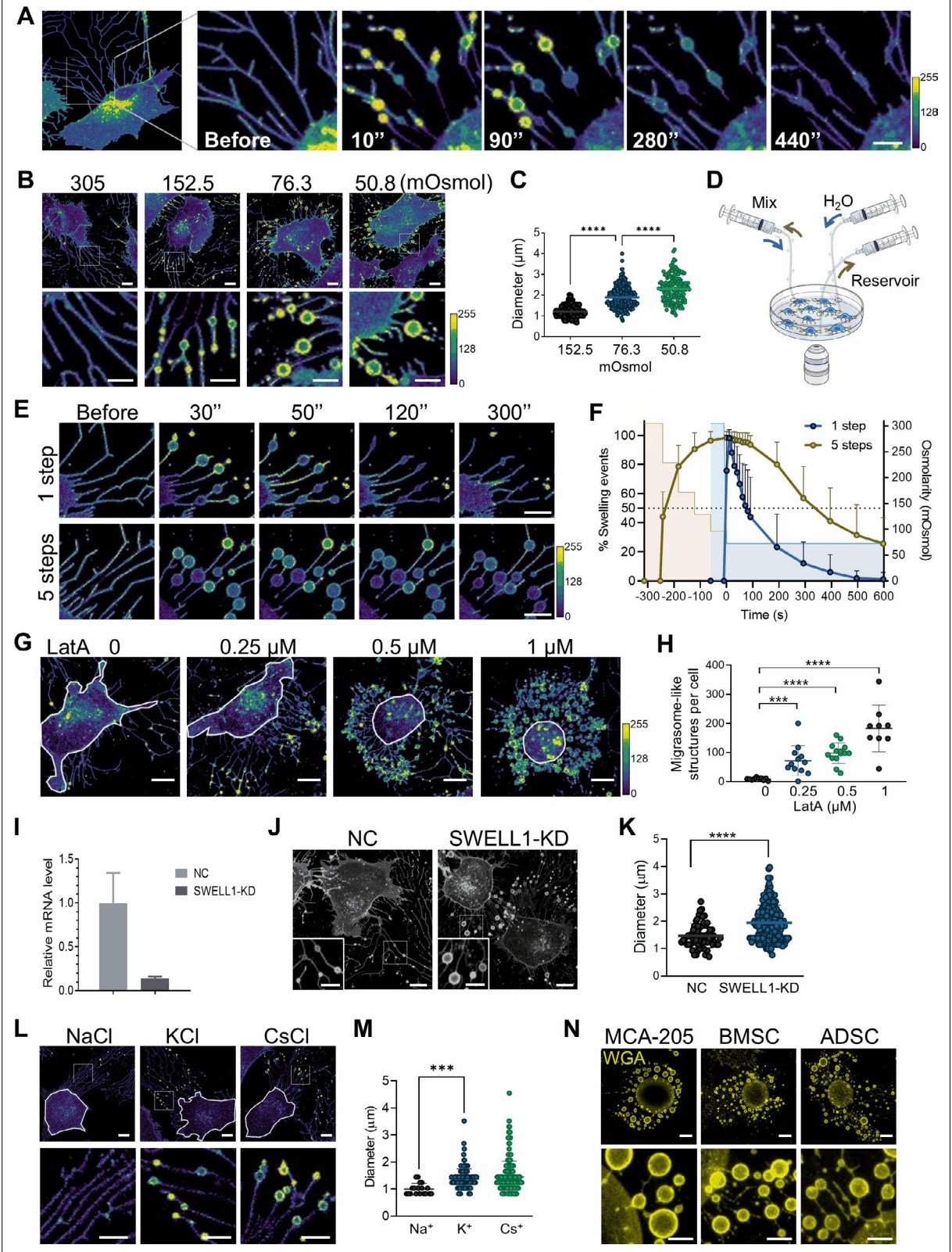

**Figure 1.** Hypotonic stimulation-induced migrasome-like structures. (**A–M**) The biogenesis of migrasome-like structures in NRK cells stably expressing Tspan4-GFP. In these images, the fluorescence intensity of Tspan4-GFP is shown in a color map scale from purple (low) to yellow (high). (**A**) Image series of the biogenesis of migrasome-like structures. Cells were treated with hypotonic Dulbecco's phosphate-buffered saline (DPBS) with an osmolality of 76.3 mOsmol and imaged using a confocal microscope. Scale bar, 10 μm. (**B**) Representative confocal images of cells treated with DPBS with various

*Figure 1 continued*

osmolarities. 305 mOsmol represents an isotonic condition in which no migrasome-like structures were observed. DPBS diluted to 152.5, 76.3, or 50.8 mOsmol was used to achieve hypotonic stimulations of different magnitude. The size of migrasome-like structures increased as the osmolarity was reduced. Scale bars, 10 µm (upper panels) and 5 µm (lower panels). (C) For each migrasome-like structure in (B), the whole lifetime, including the growth and shrinkage, was recorded by time-lapse imaging. The largest diameter reached during the lifetime was measured. The average diameter of migrasome-like structures increased significantly as the osmolarity was reduced. For hypotonic stimulation at 152.5, 76.3, or 50.8 mOsmol, n=194, 261, 165 migrasome-like structures, respectively. Data were plotted as mean ± SD. (D) Illustration of the experimental setup for real-time imaging of the induction of migrasome-like structures. (E) Image series of cells treated with hypotonic DPBS in two different approaches. The osmolarity of DPBS was reduced to 76.3 mOsmol by either one single step (upper panel) or five steps with 1-minute intervals (lower panel). For the step-wise reduction, the osmolarity was lowered by 25% in each step. Imaging time points were counted from the final stimulation step. Migrasome-like structures induced by the stepwise protocol showed significantly enhanced stability. Scale bars, 5 µm. (F) Statistical analysis of growth curves of migrasome-like structures in (E). Data were plotted as mean + SD. For single step stimulation, n=16 cells; for step-wise stimulation, n=13 cells. (G) Representative confocal images showing the effect of latrunculin A (LatA) treatment on the biogenesis of migrasome-like structures. Cells were pre-incubated with 0, 0.25, 0.5, or 1 µM LatA for 10 minutess and then treated with a five-step hypotonic stimulation as described in (E). LatA enhanced the biogenesis of migrasome-like structures in a dose-dependent manner. The white line indicates the boundary of the cell body. Scale bars, 10 µm. (H) Statistical analysis of the number of migrasome-like structures per cell in (G). Data were plotted as mean ± SD, for LatA treatment at 0, 0.25, 0.5, or 1 µM, n=10, 12, 14, or 9 cells were analyzed, respectively. (I) Relative mRNA level of SWELL1 analyzed by qPCR. SWELL1 expression was significantly reduced in SWELL1-knockdown (KD) cells compared to control cells. Data were plotted as mean + SD, n=3. (J) Representative confocal images of control or SWELL1-KD cells. Cells were treated with a five-step hypotonic stimulation at 2-minute intervals. The osmolarity was reduced by 1/6 in each step. Migrasome-like structures are shown in the inserts. Scale bars, 10 µm. Inset scale bars, 5 µm. (K) Statistical analysis of the diameter of migrasome-like structures in (J); data were plotted as mean ± SD, n=63 for control cells and 167 for SWELL1-KD cells. (L) Representative confocal images showing the effect of extracellular cations on the biogenesis of migrasome-like structures. Before stimulation, the culture medium was replaced by modified DPBS in which either $Na^+$, $K^+$, or $Cs^+$ was the only cation source. Cells were then treated with a five-step hypotonic stimulation at 2-minute intervals. The osmolarity was reduced by 1/6 in each step. Scale bars, 10 µm (upper panels) and 5 µm (lower panels). (M) Statistical analysis of the diameter of migrasome-like structures in (I); data were plotted as mean ± SD, n=18 for $Na^+$, 119 for $K^+$, and 203 for $Cs^+$. (N) Multiple primary cell types and cell lines are capable of producing eMigrasomes. Cells were stained with WGA-AF488 (Thermo, W11261) after hypotonic induction of eMigrasome with our protocol. Z-stack image series were captured and sum-slices projections were applied. Scale bars, 10 µm (upper panels) and 5 µm (lower panels). For all statistical analyses in this figure, *P* values were calculated using a two-tailed unpaired nonparametric test (Mann–Whitney test). *P* value<0.05 was considered statistically significant. \*\*\**P*<0.001; \*\*\*\**P*<0.0001.

The online version of this article includes the following video and source data for figure 1:

**Source data 1.** Original statistical data for *Figure 1C, F, H, I, K and M*.

**Figure 1—video 1.** Time-lapse movie showing the biogenesis of migrasome-like structures.

https://elifesciences.org/articles/97621/figures#fig1video1

**Figure 1—video 2.** The dynamics of migrasome-like structures induced by single-step approach.

https://elifesciences.org/articles/97621/figures#fig1video2

**Figure 1—video 3.** The dynamic of migrasome-like structures induced by a stepwise approach.

https://elifesciences.org/articles/97621/figures#fig1video3

observed in recent studies (*Zucker et al., 2025*; *Yoshikawa et al., 2024*). 10 seconds after hypotonic shock, the Tspan4 signal started to become enriched on retraction fibers. These Tspan4-enriched domains then grew into migrasome-like structures. After reaching its peak intensity, the Tspan4-GFP signal started to diffuse away from the migrasome-like structure, which was accompanied by shrinkage of the migrasome-like structure. 440 seconds after hypotonic shock, most of the hypotonic shock-induced migrasome-like structures disappeared (*Figure 1A*, *Figure 1—video 1*). We found that the size of the migrasome-like structures negatively correlated with osmolarity: the lower the osmolarity, the larger the migrasome-like structures (*Figure 1B and C*). To study the role of osmolarity in the formation of migrasome-like structures, we set up an imaging protocol which allowed us to carry out live-cell imaging while lowering the osmolarity in a step-wise manner (*Figure 1D*). We found that step-wise application of the hypotonic shock significantly increased the duration time of the migrasome-like structures. In cells undergoing five steps of hypotonic shock, 50% of the migrasome-like structures were still stable 6 minutes after the final step of hypotonic shock; in contrast, in cells undergoing one step of hypotonic shock, 50% of the migrasome-like structures shrunk back or fused with their neighbors within 1.5 minutes (*Figure 1E and F*, *Figure 1—videos 2* and *3*).

We realized that the amount of migrasome-like structures generated by hypotonic shock depends on the number of retraction fibers. We also realized that the majority of the plasma membrane can be the source of membrane for migrasome-like structures. We reasoned that if we shrink the cells by

disrupting the cytoskeleton, the shrinkage will cause the retraction of the cell edge toward the center. At the same time, the cells will be adhering to the bottom of the culture plate at various points by focal adhesion. These adhesion sites will keep the plasma membrane in place, thus serving as anchor points for generation of membrane tethers. If this scenario is true, contraction of the cell will generate large numbers of membrane tubes in a way similar to retraction fiber formation during migration. To test this hypothesis, we treated cells with different doses of latrunculin A (LatA), a reagent widely used for disruption of microfilaments, before applying the step-wise hypotonic shock. As expected, we found that treating cells with LatA caused shrinking of the cells. We also observed the massive formation of membrane tethers in the area which was occupied by the cell before shrinkage. Importantly, we observed significantly enhanced formation of migrasome-like structures from these newly formed membrane tethers in a LatA dose-dependent manner (*Figure 1G and H*).

It is well established that cells can counter a change of osmolarity with regulated volume change (*Hoffmann et al., 2009*). To test whether regulated volume change can affect the formation of migrasome-like structures, we knocked down SWELL1, a key component of the volume-regulated anion channel that maintains a constant cell volume in response to osmotic changes (*Qiu et al., 2014*; *Voss et al., 2014*). We found that knockdown of SWELL1 significantly enhanced the formation of migrasome-like structures. This result suggests that the formation of migrasome-like structures can be enhanced by reducing the capacity of cells to regulate their volume during osmolarity change (*Figure 1I–K*).

Under normal physiological conditions, the extracellular space typically contains a substantial amount of sodium. Throughout all the aforementioned experiments, the buffer employed predominantly consisted of sodium as the prevailing cation. Cations are known for their role in regulation of cell volume. Next, we tested the effect of different cations on the formation of migrasome-like structures. We reasoned that if different cations have different abilities to regulate cell volume during osmolarity change, we may be able to find an easy way to attenuate the regulated cell volume change, thus promoting the formation of migrasome-like structures. To do that, we first replaced the medium with isotonic buffers containing different cations, then we added water step-wise to reduce the osmolarity. We found that indeed different cations have different abilities to promote the generation of migrasome-like structures. Substitution of sodium with equal molar potassium or cesium significantly enhanced the generation of migrasome-like structures (*Figure 1L and M*).

Tspan4 is the key protein that promotes migrasome formation, and all the experiments described above were carried out in Tspan4-GFP-expressing cells. To test whether Tspan4 can promote the formation of migrasome-like structures, we treated mCherry- or Tspan4-mCherry-expressing cells with step-wise hypotonic shock and then observed the formation of migrasome-like structures by WGA staining, which labels migrasomes effectively (*Chen et al., 2019*). We found that indeed Tspan4-mCherry significantly enhanced the formation of migrasome-like structure (*Figure 2A and B*). Similar results held for Tspan1 and CD82, which are also migrasome-promoting tetraspanins (*Figure 2C and D*, *Figure 2—figure supplement 1*). Previously, we reported that cholesterol is essential for migrasome formation. We treated cells with methyl-beta-cyclodextrin (MβCD), which selectively extracts cholesterol from the plasma membrane. We found that the addition of MβCD rapidly destroyed the readily formed migrasome-like structures (*Figure 2E and F*). This suggests that, similar to migrasomes, the structural integrity of migrasome-like structures depends on cholesterol. Recently, we reported that the formation of migrasomes is dependent on SMS2. We found that treating cells with SMS2-IN-1, a selective inhibitor of SMS2, significantly inhibits the formation of migrasome-like structures (*Figure 2G and H*). Finally, we tested whether the formation of migrasome-like structures can occur in cells other than NRK cells. Indeed, all the cells we tested were able to support the formation of migrasome-like structures (*Figure 1N*).

Because of the mechanistic similarity between migrasomes and migrasome-like structures (*Figure 2I*), and because of the artificial nature of the procedure to generate these structures, we named these migrasome-like structures as engineered migrasomes (eMigrasomes).

Based on these results, we designed a protocol to generate and isolate eMigrasomes (eMigs) (*Figure 3A*). In this protocol, cells cultured in flasks were pretreated with high-potassium DPBS (K-DPBS) containing LatA and then the osmolarity was reduced by adding water in a step-wise manner. The flask then was subjected to moderate rotation to separate poorly adhering cell bodies and tightly adhering eMigrasomes. The supernatant containing the majority of cell bodies was discarded, and the

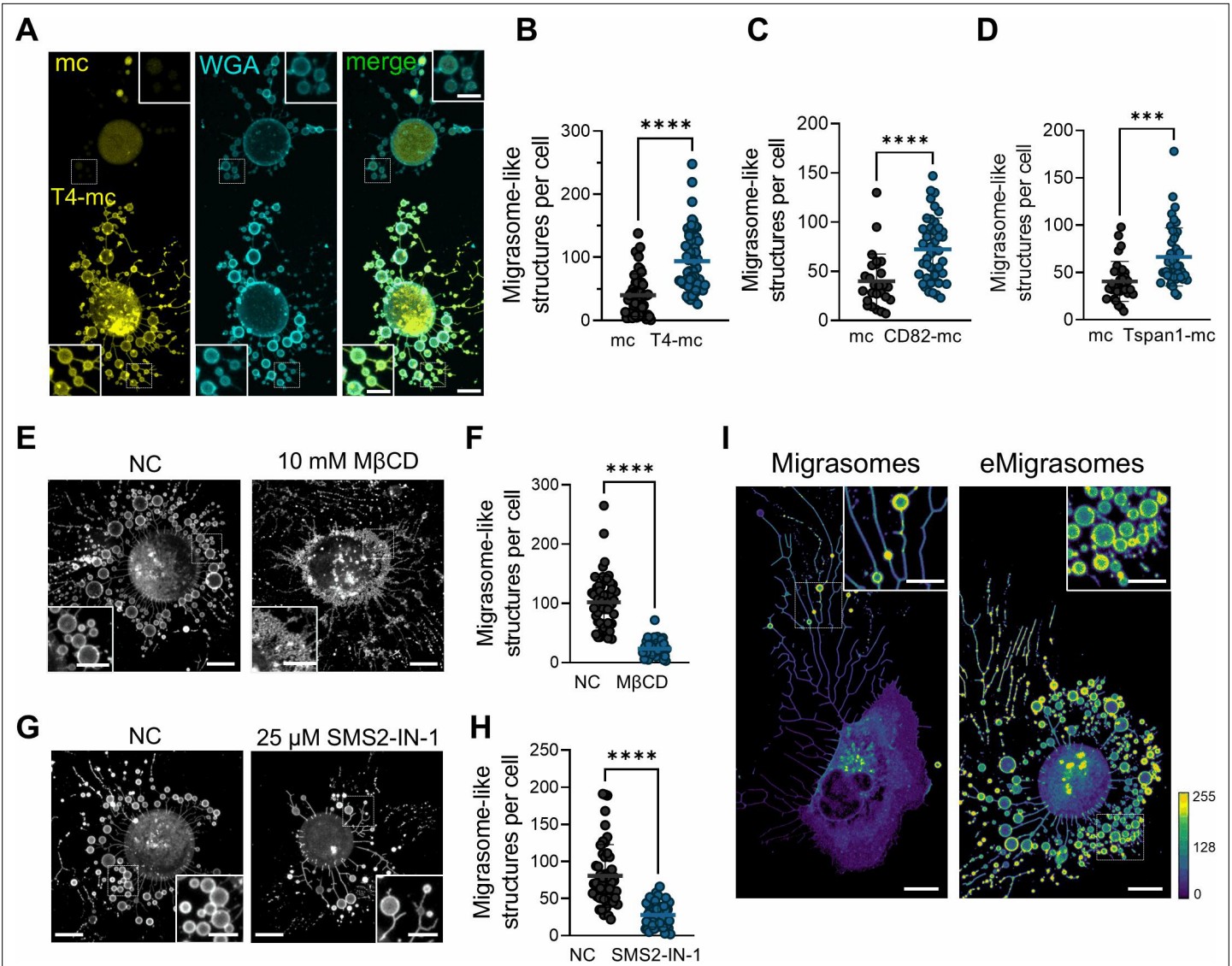

**Figure 2.** Mechanistic and morphologic similarity between migrasomes and eMigrasomes. (**A**) Representative confocal images showing the effect of Tspan4-GFP in the biogenesis of migrasome-like structures. NRK cells were transiently transfected with mCherry vector or Tspan4-mCherry. The two populations of transfected cells were mixed in a 1:1.5 ratio in a test tube and then seeded in a confocal chamber. Cells were pre-incubated with 2 µM LatA for 10 minutess and then treated with a three-step hypotonic stimulation with 2-minute intervals. In each step, the osmolarity was reduced by 1/6 (16.7%). WGA-AF647 (Thermo, W32466) was then added to stain migrasome-like structures. Z-stack images were captured for further analysis. Scale bars, 10 µm. Inset scale bars, 5 µm. (**B**) Statistical analysis of the number of migrasome-like structures per cell in NRK cells transiently transfected with mCherry vector or Tspan4-mCherry in (**A**). Data were plotted as mean ± SD, n=53, 53 cells, respectively. (**C**) Statistical analysis of the number of migrasome-like structures per cell in NRK cells transiently transfected with mCherry vector (n=26 cells) or CD82-mCherry (n=44 cells) in *Figure 2—figure supplement 1A*. Data were plotted as mean ± SD. (**D**) Statistical analysis of the number of migrasome-like structures per cell in NRK cells transiently transfected with mCherry vector (n=31 cells) or Tspan1-mCherry (n=44 cells) in *Figure 2—figure supplement 1B*. Data were plotted as mean ± SD. (**E**) Representative confocal images showing the effect of cholesterol extraction on migrasome-like structures. NRK cells stably expressing Tspan4-GFP were stimulated to generate migrasome-like structures as described in (**A**). Cells were then incubated with 10 mM MβCD or buffer supplied with an equal volume of control solvent ($H_2O$) for 30 minutes before imaging. Z-stack images were captured for further analysis. Scale bars, 10 µm. Inset scale bars, 5 µm. (**F**) Statistical analysis of the number of migrasome-like structures per cell in control cells (n=56) or cells treated with 10 mM MβCD (n=58) in (**E**). Data were plotted as mean ± SD. (**G**) Representative confocal images showing the effect of sphingomyelin depletion on the biogenesis of eMigrasomes. NRK cells stably expressing Tspan4-GFP were incubated with DMSO or 25 µM SMS2-IN-1 for 16 hours. Cells were then treated and imaged as described in (**A**). Scale bars, 10 µm. Inset scale bars, 5 µm. (**H**) Statistical analysis of the number of migrasome-like structures per cell in control cells (n=51) or cells treated with 25 µM SMS2-IN-1 (n=46) in (**G**). Data were plotted as mean ± SD. (**I**) Representative confocal images showing a cell generating natural migrasomes (left) and eMigrasomes (right). Migrasomes and eMigrasomes are morphologically similar. The fluorescence signal of Tspan4-GFP is highly enriched in both migrasomes and eMigrasomes. Scale bars, 10 µm. Inset scale bars, 5 µm. For all statistical analyses in this figure,

*Figure 2 continued on next page*

*Figure 2 continued*

*P* values were calculated using a two-tailed unpaired nonparametric test (Mann–Whitney test). *P* value<0.05 was considered statistically significant.
***P*<0.001; *****P*<0.0001.

The online version of this article includes the following source data and figure supplement(s) for figure 2:

**Source data 1.** Original statistical data for *Figure 2B, C, D, F and H*.

**Figure supplement 1.** The effect of tetraspanin expression on the formation of migrasome-like structures.

eMigrasomes attached to the bottom were harvested by pipetting. No trypsinization was applied to ensure optimal protection of the integrity of eMigrasomes. To remove remaining cell bodies, crude eMigrasomes were subjected to differential centrifugation followed by a gravity-dependent filtration through a 6 µm filter. eMigrasomes in the flowthrough were concentrated by high-speed centrifugation. This protocol generated eMigrasomes in a high yield. By microscopic examination, every individual cell robustly generates eMigrasomes (*Figure 3—video 1*). By confocal microscopy analysis, the isolated eMigrasomes are round vesicles with different sizes (*Figure 3B*). To measure the size of eMigrasomes, we carried out a confocal-based analysis using Hough circle transformation (*Figure 3B*). We found that eMigrasomes have a size range from 1.4 to 6.6 µm, with a median size of 1.6 µm (*Figure 3C*). Since negative staining might distort the shape of membrane vesicles (*Figure 3D*), we carried out cryo-EM analysis of eMigrasomes. Under cryo-EM, eMigrasomes appeared as intact round vesicles (*Figure 3E*). Notably, the eMigrasomes were not contaminated with significant amounts of intracellular membranes, as western blotting of purified eMigrasomes showed very little contamination from other organelles (*Figure 3F*).

Previously, we reported that migrasomes can become leaky before rupture. We wondered whether eMigrasomes can also become leaky. To test this, we added eMigrasomes into isolation buffer loaded with Cy5 and 40 kDa dextran-TMR. At RT, eMigrasomes rapidly become leaky to Cy5, a fluorescent dye that does not pass intact membranes (*Figure 3G*). During the 48 hours after preparation, eMigrasomes gradually become leaky to 40 kDa dextran-TMR, which mimics the size of a normal protein (*Figure 3G and H*).

Next, we tested the stability of eMigrasomes at RT. Surprisingly, eMigrasomes are highly stable. Even after 14 days at RT, the morphology of eMigrasomes did not change significantly, and the number of eMigrasomes was only slightly reduced (*Figure 3I and J*). However, if we treated isolated eMigrasomes with MβCD, most of them deformed and ruptured within 30 minutes (*Figure 3K*). This suggests a crucial contribution of cholesterol to the stability of isolated eMigrasomes.

The stable nature of eMigrasomes prompted us to explore the possibility of using eMigrasomes as carriers for delivery of proteins (*Figure 4A*). We found that membrane proteins (e.g., the cell surface receptor PD1) can be easily loaded onto eMigrasomes by simply overexpressing these proteins in cells (*Figure 4B*). To load cytosolic proteins, we fused them with the transmembrane domain followed by the polybasic tail of syntaxin 2 (STX2). This allowed us to successfully load the cytosolic protein OVA onto the plasma membrane and thus onto eMigrasomes, referred to as mOVA for membrane-tethered OVA (*Figure 4C*).

## An eMigrasome-based vaccine

Next, we explored the potential of eMigrasomes as an antigen carrier for vaccines. First, we employed OVA, a well-established model antigen, to evaluate eMigrasomes as a platform for antigen delivery. We used imaging and immunoblotting to confirm the presence of the mOVA-mCherry protein (*Figure 4D and E*). It is worth noting that the mOVA antigen was highly enriched in isolated eMigrasomes compared to the host cells (*Figure 4E*). We immunized mice with OVA-loaded eMigrasomes (eM-OVA) via different routes and found that intravenous injection resulted in the highest IgG antibody titer (*Figure 4F*). eM-OVA induced the antigen-specific IgG response in a dose-dependent manner (*Figure 4G*). The IgG response induced by eM-OVA at 20 µg/mouse was comparable to traditional Alum/OVA at 50 µg/mouse (*Figure 4H*). Together, these data suggest that eM-OVA elicits a strong IgG response compared to traditional alum-based immunization.

IgG is the most abundant immunoglobulin in human and mouse, with four different subtypes: IgG1, IgG2, IgG3, and IgG4 in human, and IgG1, IgG2a/c, IgG2b, IgG3 in mouse (*Martin et al., 1998*). Different IgG subtypes are highly conserved, but each has its unique immunological functions, for

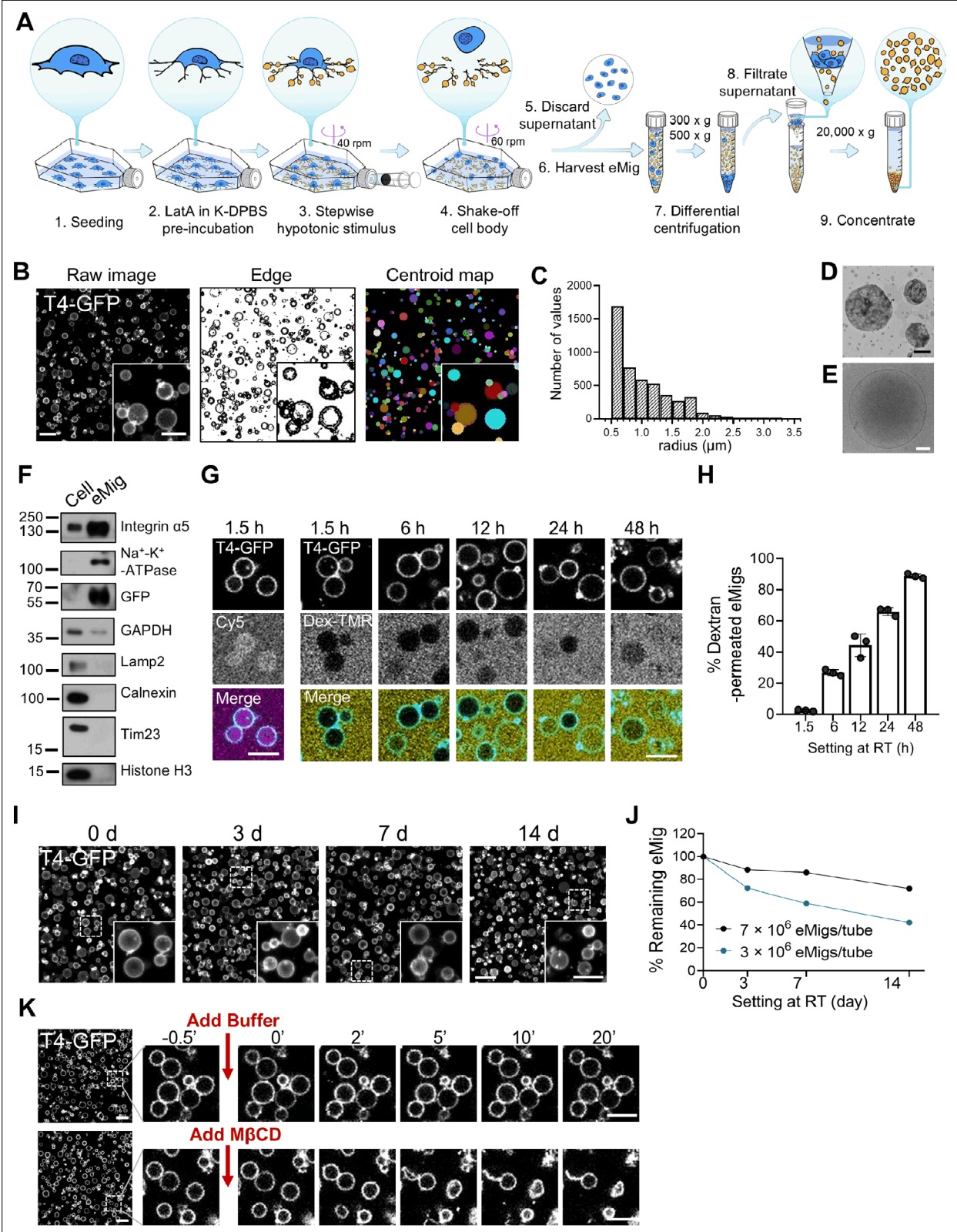

**Figure 3.** Isolation and characterization of eMigrasome. NRK cells stably expressing Tspan4-GFP were used in all experiments in this figure if not otherwise specified. (**A**) Schematic illustration showing the process of eMigrasome induction, isolation, and purification. (**B**) Confocal image (left), threshold edge (middle), and centroid map (right) of purified eMigrasomes. Image processing and analysis were performed using ImageJ. The Hough circle transform plugin was applied to recognize and transform thresholded edges into binned objects representing individual eMigrasomes. Scale

*Figure 3 continued on next page*

*Figure 3 continued*

bar, 10 µm. Inset scale bar, 5 µm. (**C**) Statistical analysis of the radius of purified eMigrasomes. Measurement was performed using the map generated by Hough circle transformation analysis. 4725 particles were analyzed and the data were binned to plot the distribution of eMigrasomes radius. (**D**) TEM micrograph of negatively stained purified eMigrasomes. Scale bar, 1 µm. (**E**) Cryo-EM micrograph of purified eMigrasomes. Scale bar, 200 nm. (**F**) Western blot showing the protein level of several markers in cell bodies (Cell) and eMigrasomes (eMig). An equal amount of protein was loaded in each lane. (**G**) Representative time-lapse confocal images showing the high permeability of eMigrasomes to Cy5 at 1.5 hours post purification (left) and the gradual increase in the permeability of eMigrasomes to 40 kDa dextran-TMR (right). Scale bars, 5 µm. (**H**) Statistical analysis of the percentage of eMigrasomes (eMigs) that were permeable to 40 kDa dextran-TMR at the indicated time points. For each time point, eMigrasomes from three different views were analyzed (data were plotted as mean ± SD, n=3). From left to right, 336, 493, 812, 830, and 631 eMigrasomes were analyzed. (**I**) Representative confocal images of eMigrasomes after sitting at room temperature for 0, 3, 7, or 14 days. Aliquots of eMigrasomes were stored in EP tubes as pellets at room temperature for the indicated time, then resuspended and dropped into a confocal chamber before imaging. Z-stack image series were captured and sum-slices projections were applied. Scale bar, 10 µm. Inset scale bar, 5 µm. (**J**) Number of eMigrasomes after storage at room temperature for 0, 3, 7, or 14 days. Aliquots of eMigrasomes ($7 \times 10^6$ eMigrasomes per tube in black, $3 \times 10^6$ eMigrasomes per tube in green) were stored in EP tubes as pellets at room temperature for the indicated time, then resuspended and stained with WGA561 before counting by FACS. (**K**) Time-lapse image series showing purified eMigrasomes treated with 10 mM MβCD or control buffer. A 10 µl drop of concentrated eMigrasomes was settled in a confocal chamber, then sealed and maintained at 37°C during imaging. Buffer containing 10 mM MβCD or control solvent was added to the drop using the equipment illustrated in *Figure 1D*. Scale bars, 10 µm (left panels) and 5 µm (right panels).

The online version of this article includes the following video and source data for figure 3:

**Source data 1.** Original western blots for *Figure 3F*, indicating the relevant bands and treatments.

**Source data 2.** Original files for western blot analysis displayed in *Figure 3F*.

**Source data 3.** Original statistical data for *Figure 3C, H and J*.

**Figure 3—video 1.** 4D time-lapse movie showing the biogenesis of eMigrasomes.

https://elifesciences.org/articles/97621/figures#fig3video1

example, acting through different Fc-gamma receptors (FcγRs) or binding to complement (*Vidarsson et al., 2014*). Thus, we characterized the type of IgG induced by eM-OVA in mice and compared it to that induced by Alum/OVA. The IgG response to Alum/OVA was dominated by IgG1, consistent with previous publications (*Comoy et al., 1997*; *Hogenesch, 2012*). Quite differently, eM-OVA induced an even distribution of IgG subtypes, including IgG1, IgG2b, IgG2c, and IgG3 (*Figure 4I*). Usually, the ratio between IgG1 and IgG2a/c indicates a Th1 or Th2-type humoral immune response. Thus, eM-OVA immunization induces a balance of Th1/Th2 immune responses.

Next, we assessed the stability of the immunogenicity of eM-OVA over a period of 14 days at RT. For this purpose, we placed the purified eM-OVA in a test tube at RT without adding any reagents to inhibit protein degradation. After 14 days of RT storage, the amount of intact OVA was roughly the same as in freshly purified eM-OVA, and the IgG response induced by eM-OVA remained unchanged during this period (*Figure 4J–L*).

## An eMigrasome-based vaccine induces a strong humoral protective response against SARS-CoV-2

We next explored eMigrasomes as a platform to carry the SARS-CoV-2 Spike protein (S protein). To prevent the S protein from being broken down by proteases, we mutated the furin site of the S protein. MCA-205 cells were used to express the S protein with an mCherry tag. After isolating the eMigrasomes, we used imaging to confirm the presence of the Spike protein (*Figure 5A*). To assess the antigen integrity, we performed immunoblotting using antibodies against both S1 and mCherry. Two distinct bands were observed: one at the expected molecular weight of the S-mCherry fusion protein, and a higher molecular weight band that may represent oligomerized or higher-order forms of the Spike protein (*Figure 5B*). Furthermore, we performed confocal microscopy using a monoclonal antibody against Spike. Co-localization analysis revealed strong overlap between the mCherry fluorescence and anti-Spike staining, confirming the proper presentation and surface localization of intact S-mCherry fusion protein on eMigrasomes (*Figure 5C*). These results confirm the structural integrity and antigenic fidelity of the Spike protein expressed on eMigrasomes.

Immunization of WT C57BL/6J mice with the Spike-loaded eMigrasomes (eM-S) resulted in a strong antibody response against the S protein, which was further improved by a second shot (*Figure 5D and F*). To assess the neutralizing capacity of the antisera provoked by eM-S, we utilized a replication-competent, infectious VSV chimera incorporated with the SARS-CoV-2 spike protein for a neutralization

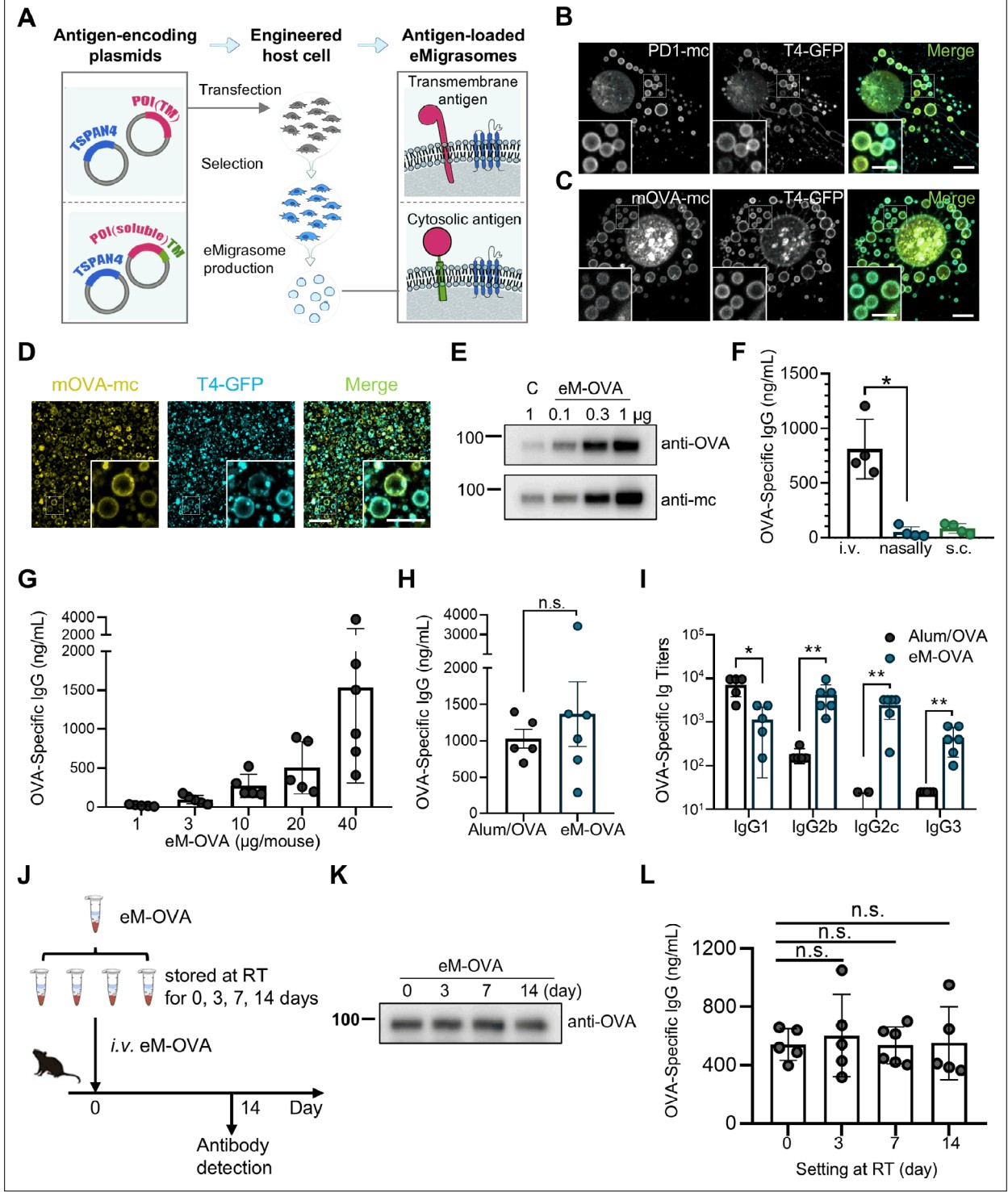

**Figure 4.** eMigrasomes as an antigen carrier for vaccination. (**A**) Schematic illustration of strategies for loading proteins of interest (POI) onto eMigrasomes. The protocol includes the construction of antigen-encoding plasmids, the construction of engineered cell lines stably expressing the antigens, and the production of antigen-loaded eMigrasomes. Membrane proteins, which already carry a transmembrane (TM) domain, are overexpressed in cells (top). Cytosolic proteins are membrane tethered by tagging with the TM sequence and polybasic tail of STX2 (bottom). Cells are then treated with hypotonic buffer to enrich the POI on the surface of eMigrasomes. (**B**) Representative confocal image of a cell expressing PD1-mCherry and Tspan4-GFP. The transmembrane protein PD1-mCherry was loaded onto eMigrasomes as shown in the top part of (**A**). Scale bars, 10 µm. Inset scale bars, 5 µm. (**C**) Representative confocal image of a cell expressing membrane-tethered OVA-mCherry (mOVA-mc) and Tspan4-GFP. To create an extracellular membrane-tethered form of OVA (mOVA), the sequence of OVA was fused to the C-terminus of a truncated form of mouse STX2, in

*Figure 4 continued on next page*

*Figure 4 continued*

which only the transmembrane region and a polybasic tail remained. The mCherry tag was fused to the C-terminus of OVA to trace the localization of this fusion protein. The soluble protein OVA was loaded onto the membrane of eMigrasomes, as shown in the bottom part of (**A**). Scale bars, 10 µm. Inset scale bars, 5 µm. (**D**) Representative confocal image of eMigrasomes isolated from MCA-205 cells stably expressing Tspan4-GFP and mOVA-mCherry (eM-OVA). Scale bar, 10 µm. Inset scale bar, 5 µm. (**E**) Western blot showing the amount of full-length mOVA-mCherry protein in host cell and eM-OVA. Cell lysate (**C**) containing 1 µg total protein and purified eM-OVA samples containing 0.1, 0.3, or 1 µg total protein were loaded. The protein-immobilized PVDF membrane was firstly incubated with anti-OVA antibody and then stripped and re-blotted with anti-mCherry antibody. The antigen mOVA-mCherry was highly enriched in eMigrasomes compared to host cells. MCA-205 cells stably expressing Tspan4-GFP and mOVA-mCherry were used for all experiments in the rest of this figure if not otherwise specified. (**F**) Amount of OVA-specific IgG in mouse serum on day 14 after intravenous (i.v), nasal, or subcutaneous (s.c) immunization with eM-OVA (20 µg/mouse). OVA-specific IgG was quantified by ELISA. Data were plotted as mean ± SD, n=4 mice were analyzed for each group. (**G**) ELISA quantification of OVA-specific IgG in sera from wild-type (WT) mice on day 14 after tail intravenous immunization with eM-OVA at the indicated dose. Data were plotted as mean ± SD. For 1, 3, 10, 20 µg/mouse eM-OVA immunization, n=5 mice; for 40 µg/mouse eM-OVA immunization, n=6. (**H**) ELISA quantification of OVA-specific IgG in sera from WT mice on day 14 after tail intravenous immunization with eM-OVA (20 µg/mouse, n=6 mice) or intraperitoneal immunization with Alum/OVA (n=5 mice). Data were plotted as mean ± SD. (**I**) Titer analysis of OVA-specific IgG1, IgG2b, IgG2c, and IgG3 in the sera from mice immunized with eM-OVA (20 µg/mouse, n=6 mice) or Alum/OVA (n=5 mice). Serum samples were collected on day 14. Each dot represents an individual serum sample. Data were plotted as mean ± SD. (**J**) Illustration of the experimental setup for assaying the stability of eM-OVA. (**K**) Immunoblotting analysis of the amount of OVA protein in samples of eM-OVA that were left at room temperature for 0, 3, 7, or 14 days. 2 µg protein was loaded in each lane. (**L**) ELISA quantification of OVA-specific IgG in sera from WT mice on day 14 after tail intravenous immunization with eM-OVA stored at room temperature for 0 days (D0, n=5 mice), 3 days (D3, n=5 mice), 7 days (D7, n=6 mice), or 14 days (D14, n=5 mice). Data were plotted as mean ± SD. 20 µg eM-OVA was injected per mouse. For all statistical analyses in this figure, P values were calculated using a two-tailed unpaired nonparametric test (Mann–Whitney test). P value<0.05 was considered statistically significant. n.s. P>0.05; *P<0.05; **P<0.01.

The online version of this article includes the following source data for figure 4:

**Source data 1.** Original western blots for *Figure 4E and K*, indicating the relevant bands and treatments.

**Source data 2.** Original files for western blot analysis displayed in *Figure 4E and K*.

**Source data 3.** Original statistical data for *Figure 4F, G, H, I and L*.

test (*Figure 5E*), similar to the previously reported system (*Case et al., 2020b*; *Case et al., 2020a*). This genetically altered VSV chimera virus features the SARS-CoV-2 spike protein, substituting its native surface glycoprotein (G), making the VSV chimera reliant on the SARS-CoV-2 spike protein for cellular entry. The Venus-based fluorescence reporter system offers high sensitivity. The neutralizing power of antisera against the SARS-CoV-2 spike protein was assessed by calculating the percentage of Venus-positive infected cells when treated with serum versus mock controls. The antisera, stimulated by an initial immunization with eM-S, neutralized the recombinant VSV-Venus-SARS-CoV2 up to a dilution titer of approximately 300 to achieve 50% neutralization ($NT_{50}$). A secondary booster immunization further enhanced the neutralization capability with an $NT_{50}$ up to 4000 (*Figure 5G and H*). Together, these results indicate that antigen-carrying eMigrasomes can induce a strong humoral protective response against SARS-CoV-2.

## Discussion

In this article, we describe a method to rapidly generate eMigrasomes from cultured mammalian cells. We developed a simple method to load membrane or cytosolic proteins onto eMigrasome. Using OVA as a model antigen, we demonstrate that eMigrasomes are a highly effective, temperature-stable vaccine platform which can elicit antibody response with a very small amount of antigen. Finally, we show that eMigrasomes can be used to generate effective vaccines against SARS-CoV-2. Collectively, our study provides the proof of concept for developing eMigrasome-based vaccines.

Previously, migrasomes have been defined as 'migration-dependent" vesicles. In this study, we demonstrated that migrasome-like structures can be induced through the relative movement of the cell edge in a migration-independent manner. Notably, eMigrasomes exhibit conserved genetic and morphological features compared to natural migrasomes, providing strong evidence for this concept. Accordant with our study, a recent investigation has revealed that cell shrinkage induced by bacterial toxins can also trigger migrasome formation (*Li et al., 2024*).

In this study, we demonstrate the efficiency of eMigrasome-based vaccines using a model antigen or an antigen from a coronavirus. The benefit of using a model antigen is that there are readily available experimental systems. The benefit of using a well-characterized virus antigen is that it allows

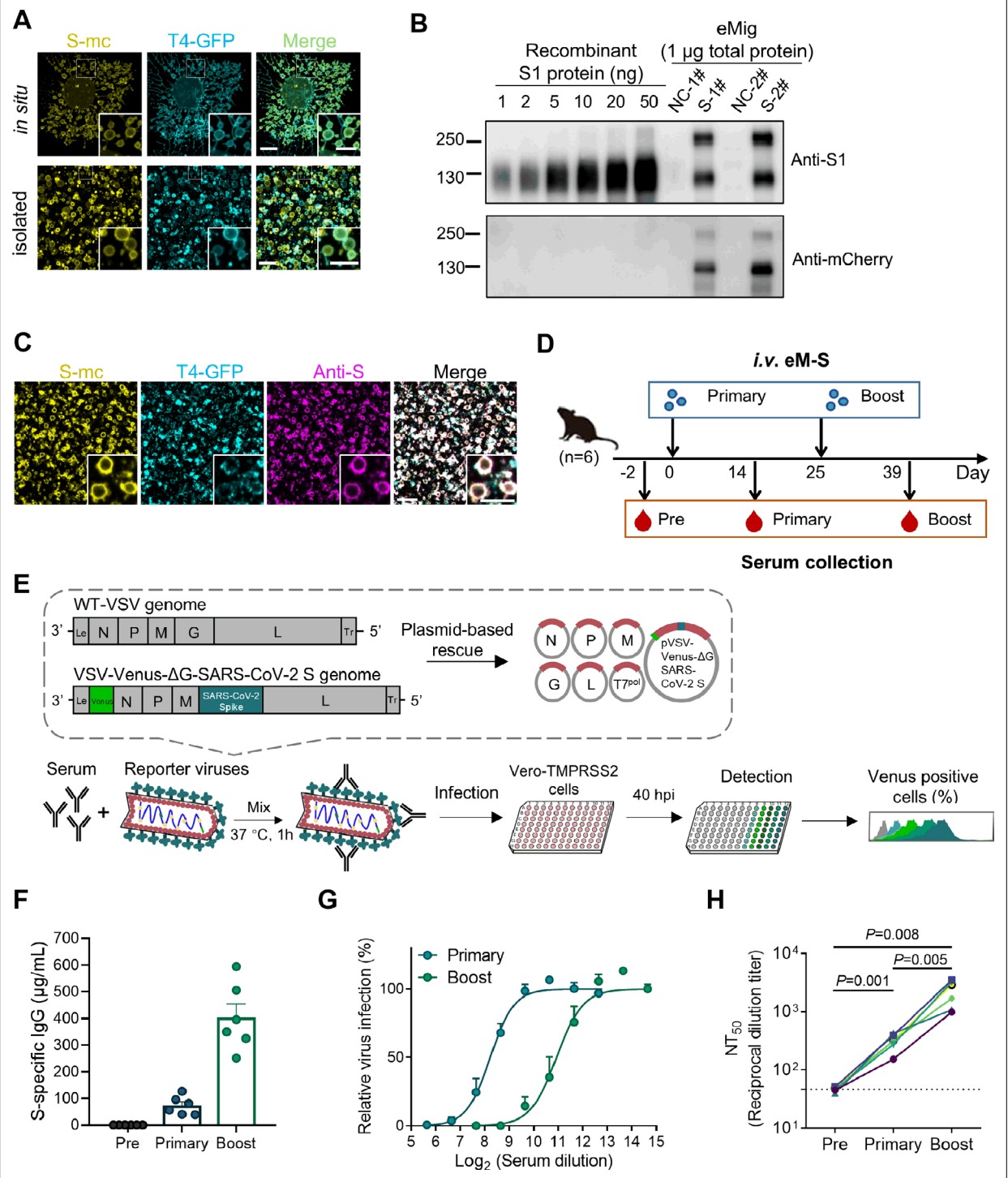

**Figure 5.** An eMigrasome-based vaccine induces a strong humoral protective response against SARS-CoV-2. (**A**) Representative confocal images showing the presence of Spike-mCherry (S-mc) in engineered cells and isolated eMigrasomes. Scale bars, 10 μm. Inset scale bars, 5 μm. (**B**) Western blot showing the amount of full-length Spike-mCherry protein in eM-S. A titration (1, 2, 5, 10, 20, or 50 ng) of recombinant spike protein was loaded as standards. Purified eM-NC or eM-S samples containing 1 μg total protein were loaded. (**C**) Representative confocal images showing the presence of integral spike protein in isolated eMigrasomes. Scale bars, 10 μm. Inset scale bars, 5 μm. (**D**) Schematic diagram of the experimental procedure for immunization with Spike-loaded eMigrasomes (eM-S) and collection of serum. (**E**) Illustration of the rVSV-venus-SARS-CoV-2 system. VSV-Venus-SARS-CoV-2 was mixed with vaccinated mice sera. Vero-TMPRSS2 cells were infected with the reporter virus/serum mixture with an MOI of 0.01. 40 hours post-infection, the Venus-positive infected cells were quantified to estimate the NT$_{50}$ value for each serum. (**F**) Spike-specific IgG was quantified in WT mice immunized with eM-S (20 μg/mouse, i.v.) at different time points. Data were plotted as mean ± SD. Each symbol represents one individual animal,

*Figure 5 continued on next page*

*Figure 5 continued*

n=6 mice were analyzed. (**G**) Neutralization curves are presented for sera from primary-vaccination and boost-vaccination. Data were plotted as mean + SEM, n=6 mice were analyzed. Nonlinear regression was performed using the equation for the normalized response versus the inhibitor, incorporating a variable slope. (**H**) NT$_{50}$ of individual mouse in vaccinated groups was compared by *P*-value (paired *t*-test) and is indicated. Dotted lines represent assay limits of detection. Each line represents an individual mouse, n=6 mice were analyzed.

The online version of this article includes the following source data for figure 5:

**Source data 1.** Original western blots for *Figure 5B*, indicating the relevant bands and treatments.

**Source data 2.** Original files for western blot analysis displayed in *Figure 5B*.

**Source data 3.** Original statistical data for *Figure 5F–H*.

us to reliably test the efficacy of our vaccine platform in a realistic setting. It should be noted that both a signal peptide and a transmembrane domain are necessary for proper antigen presentation on eMigrasomes. For antigens that do not naturally contain these two features, a non-native signal peptide or an artificial transmembrane domain should be engineered into the coding sequence of the antigen. We are aware that due to the experimental nature of this work, it is unlikely that the eMigrasome platform will be used to generate mainstream preventive vaccines in the foreseeable future. However, there is a very real possibility of developing eMigrasomes into therapeutic vaccines against cancer or other diseases with unmet medical needs, or into preventive vaccines against pathogens for which the current vaccine platforms do not work. Our investigation regarding the biophysical and cellular mechanisms of eMigrasome formation makes it possible for further optimizing eMigrasome generation in a rational manner.

In this study, we demonstrate that eMigrasomes can be an effective vaccine platform. In addition, we speculate that eMigrasomes may also emerge as a versatile delivery system for a range of applications. The high rigidity and self-repair capacity of migrasomes, which is caused by enrichment of tetraspanins and cholesterol, makes migrasomes highly stable, and thus suitable as delivery carriers in various in vivo settings. More importantly, the physiological roles of migrasomes indicate that they have naturally evolved as carriers for delivering materials and information in vivo. Our previous work showed that during embryonic development, migrasomes, which are highly enriched with signaling molecules, such as chemokines, growth factors, and morphogens, are deposited at various spatially defined locations, where they act as sustained-release capsules to liberate the signaling molecules. In this way, migrasomes affect multiple aspects of embryonic development, including organ morphogenesis and angiogenesis (*Jiang et al., 2019*; *Zhang et al., 2022*). Additionally, in cultured fibroblast cells, migrasomes are enriched with a selected set of full-length, translationally competent mRNAs. When these mRNA-enriched migrasomes are taken up by neighboring cells, the mRNA can escape the endo-lysosome system of recipient cells and be translated into protein, thus modifying the behaviors of the recipient cells (*Zhu et al., 2021*). Finally, in cells experiencing mild mitochondrial damage, the damaged mitochondria can be selectively transported into migrasomes and then evicted from the cell in a process named mitocytosis (*Jiao et al., 2021*). In summary, multiple types of cargos, including materials and information, can be enriched in migrasomes under different settings, and migrasomes can be deposited at spatially defined locations in diverse biological settings to affect a broad range of biological processes, including cell-cell communication. Since eMigrasomes capture certain key features of migrasomes, it is our speculation that we may be able to develop eMigrasomes into a delivery system for diverse cargo types including nucleic acids, proteins, small molecules, and even organelles.

# Materials and methods

**Key resources table**

| Reagent type (species) or resource | Designation | Source or reference | Identifiers | Additional information |
| --- | --- | --- | --- | --- |
| Cell line (*Rattus norvegicus*) | NRK | ATCC | CRL-6509 | |
| Cell line (*Homo sapiens*) | HEK 293T | ATCC | CRL-3216 | |

*Continued on next page*

*Continued*

| Reagent type (species) or resource | Designation | Source or reference | Identifiers | Additional information |
|---|---|---|---|---|
| Cell line (*Mus musculus*) | MCA-205 | | | STR profiling, authenticated by Procell Life Science & Technology |
| Cell line (*Cercopithecus aethiops*) | Vero | ATCC | CCL-81 | |
| Cell line (*H. sapiens*) | ADSC | Procell | CP-H202 | Human Adipose-derived Mesenchymal Stem Cells |
| Cell line (*H. sapiens*) | BMSC | Procell | CP-H166 | Human Bone Marrow Mesenchymal Stem Cells |
| Antibody | Anti-ovalbumin (mouse monoclonal) | Santa Cruz | sc-80589 | ELISA (1:4000– 1:512,000), WB (1:1000) |
| Antibody | Anti-Spike (mouse monoclonal) | Sino Biological | 40591-MM42 | ELISA (1:5000– 1:640,000), WB (1:1000) |
| Antibody | Anti-Spike (chimeric MAb) | Sino Biological | 40150-D001 | IF (1:100) |
| Recombinant DNA reagent | pB-mOVA-mCherry (plasmid) | This paper | | pB-mCherry derived plasmid |
| Recombinant DNA reagent | pB-S-mCherry (plasmid) | This paper | | pB-mCherry derived plasmid |
| Recombinant DNA reagent | pB-Tspan4-GFP (plasmid) | *Jiao et al., 2021* | | pB-GFP derived plasmid |
| Peptide, recombinant protein | Ovalbumin | Sigma | A5503 | |
| Chemical compound, drug | Latrunculin A | Cayman | 10010630 | 0.25–2 µM |
| Other | Imject Alum | Thermo Fisher | 77161 | |
| Other | Fibronectin | Sigma | F0895 | 10 µg/ml for imaging; 4 µg/ml for eMigrasome preparation |
| Other | 40 kDa Dextran-TMR | Thermo Fisher | D1842 | (25 µg/ml) |

## Molecular cloning

For the construction of pB-Tspan4-GFP, pB-PD1-mCherry, pB-S-mCherry, pmCherry-N1-Tspan4, pmCherry-N1-CD82, and pmCherry-N1-Tspan1, the coding sequence of the corresponding genes was amplified by PCR and subsequently inserted into the vector using a One Step Cloning Kit (Vazyme, C112). For the construction of pB-mOVA-mCherry, the sequence spanning amino acids 252–289 of syntatin-2 (STX-2), which incorporates the transmembrane domain along with the polybasic tail, was amplified from mouse cDNA by polymerase chain reaction (NEB, M0492L) using primers below:

> fwd: 5'-atgaagaaagccatcaaataccagagc-3';
> rev: 5'-gcaccgatggagcccatcgagcctttgccaaccgacaagccaatg-3'.

The sequence of OVA was amplified from the plasmid using primers below:

> fwd: 5'-atgggctccatcggtgc-3';
> rev: 5'-aggggaaacacatctgccaaag-3'.

Using STX2$_{252-289}$ and OVA as templates, the membrane-tethered mOVA, in which STX2$_{252-289}$ sequence is tagged to OVA sequence, was further amplified using the primers below:

> fwd: 5'-tcagatctcgagctcaagcttatgaagaaagccatcaaataccagagc-3'
> rev: 5'-tggtggcgaccggtggatcccggaggaagaacactaaggcagcaaaagagaag-3'

The fragment was inserted into pmCherry-N1 vector using a One Step Cloning Kit (Vazyme, C112).

## Cell culture

Cell lines used in this study include NRK (ATCC CRL-6509), HEK 293T (ATCC CRL-3216), MCA-205 (STR profiling, authenticated by Procell Life Science & Technology), Vero (ATCC CCL-81), ADSC (Procell CP-H202), and BMSC (Procell CP-H166). All the cell lines were routinely tested for mycoplasma, and only when tested negative were used for the experiments. No cell lines used in this study were found in the database of commonly misidentified cell lines that is maintained by ICLAC.

Cells were cultured at 37°C with 5% $CO_2$. NRK cells and HEK 293T cells were grown in DMEM (Gibco, C11995500BT) supplemented with 10% (v/v) FBS (Biological Industries), 1% (v/v) GlutaMAX (Gibco, 35050-061), and 1% (v/v) penicillin–streptomycin. MCA-205 cells were grown in RPMI 1640 (Gibco, C11875500BT) supplemented with 10% (v/v) FBS, 1% (v/v) GlutaMAX, and 1% (v/v) penicillin–streptomycin. Vero-TMPRSS2 cells were maintained in DMEM supplemented with 10% (v/v) FBS and 50 IU/ml penicillin–streptomycin. ADSC cells and BMSC cells were grown in stem cell serum-free medium (CytoNiche, RMZ112).

## Cell transfection and cell line development

For NRK cells, transfection was performed using electroporation (AMAXA, Nucleofector). For MCA-205 cells, transfection was performed using Lipofectamine 3000 transfection reagent (Thermo Fisher Scientific, L3000015). In establishing stable cell lines, the transfected cells underwent initial selection with Hygromycin B (Roche, 10843555001) and were subsequently sorted into single colonies in 96-well plates via flow cytometry.

## Imaging

Confocal imaging of live cells and isolated eMigrasomes was performed using a Nikon A1HD25 laser scanning confocal microscope. Cells were cultured in a confocal chamber (Cellvis, D35-20-1-N or D35C4-20-1.5-N) that had been pre-treated with 10 μg/ml fibronectin (F0895) and allowed to proliferate overnight (14–18 hours). For visualizing eMigrasomes, a small region in the confocal chamber was coated with a 10 μl drop of 10 μg/ml fibronectin. Following aspiration of the fibronectin, a 10 μl drop of isolated eMigrasomes was dispensed into the coated area and left to settle for a minimum of 1 hour prior to imaging. To prevent evaporation, the chamber was sealed using parafilm. For immunofluorescence assay, eMigrasomes were settled at 4°C overnight and then fixed by 2% PFA. Nonspecific binding was blocked by incubating with 10% (v/v) FBS. eMigrasomes were then stained with a spike primary antibody (40150-D001, SinoBiological) for 1 hour at RT, triple washed with PBS, and then stained with AlexaFluor647-conjugated secondary antibody. Samples were triple washed with PBS before imaging.

## Real-time hypotonic stimulation

To observe the dynamics of migrasome-like structures and eMigrasomes, a custom-made buffer displacement device was utilized (refer to *Figure 1D*). Essentially, three apertures were created on the lid of a confocal chamber to perfectly accommodate a silicon microinjection tube with an external diameter of 1.9 mm. The silicon microinjection tube was securely connected to a 1 ml syringe after the removal of its sharp needle.

For a single-step stimulation process, two syringes were employed. One was kept empty, while the other was filled with the desired hypotonic buffer. During time-lapse imaging, the reservoir solution was aspirated using the empty syringe, and the hypotonic buffer was simultaneously injected from the other syringe.

In the case of stepwise stimulation, three syringes were deployed. The first was loaded with the initial buffer from the reservoir and was used to withdraw the reservoir solution. The second syringe contained deionized water ($ddH_2O$) and was used to add a specific volume of water at each step. The third syringe, which was empty, was used to ensure thorough mixing of the reservoir solution after each stimulation event.

## Drug treatment

For SMS2 inhibition, 30 minutes post seeding of the cells, the culture medium was gently replaced with fresh medium containing either 25 μM SMS2-IN-1 (MCE, HY-102041) or 0.25 (v/v) % DMSO. Cells were treated for 16 hours prior to imaging.

For cholesterol extraction from in situ or isolated eMigrasomes, h-KDPBS containing 10 mM MβCD (Sigma, 332615) or 10 (v/v) % $ddH_2O$ was applied. For in situ eMigrasomes, z-stack images were collected after 30 minutes incubation at 37°C. For isolated eMigrasomes, buffer containing MβCD or $ddH_2O$ was applied using the real-time stimulation device described above. A 20-minute time-lapse imaging session was performed to capture the process of morphological changes in eMigrasomes.

## Permeability assay

Isolated eMigrasomes were diluted in h-KPBS-BSA, containing 5 µg/ml of Cy5 and 25 µg/ml of 40 kDa Dextran-TMR (D1842). A droplet of the eMigrasome solution was prepared and imaged following the procedure described previously.

## Ion replacement assay

The standard formulation of DPBS is 138 mM NaCl, 8.1 mM $Na_2HPO_4$, 2.7 mM KCl, 1.5 mM $KH_2PO_4$. In the ion replacement assay, as referenced in *Figure 1L and M*, the 138 mM NaCl in DPBS was substituted with equal molar KCl or CsCl. The other three components were not changed.

The K-DPBS formulation utilized for eMigrasome production contains 140.6 mM KCl, 1.5 mM $KH_2PO_4$, 8.1 mM $K_2HPO_4$.

## Induction and purification of eMigrasomes

It is important to underscore that several critical parameters, such as the seeding density, the working concentration of LatA, and the incubation period, along with the hypotonic gradient, should be individually optimized for each cell type. The quantities of cells and reagents utilized should be modulated according to the flask size (examples are provided in the following table).

| Cell type | NRK | MCA-205 | ADCS | BMSC |
|---|---|---|---|---|
| Seeding density (cells/cm$^2$) | $1.5 \times 10^4$ | $3 \times 10^4$ | $2 \times 10^4$ | $2 \times 10^4$ |
| Latrunculin A concentration (µM) | 2 | 2 | 0.6 | 0.6 |
| Latrunculin A incubation time (minutes) | 10 | 45 | 20 | 20 |
| Reduced osmolarity per step | 1/6 | 1/4 | 1/6 | 1/6 |

In the following method, the purification of eMigrasomes from NRK cells cultured in a single T75 flask (NEST, 708003) is used as a representative example. Three major steps include cell preparation, eMigrasome induction, and purification.

### Cell preparation

A T75 flask was coated with 4 µg/ml of fibronectin. $1 \times 10^6$ cells were seeded into the pre-coated flask, allowed to grow for 14–16 hours.

### eMigrasome induction

Cortical actin disruption was achieved by discarding the culture medium and rinsing the cells once with PBS. Then, 7.5 ml of K-DPBS containing 2 µM of LatA was added to the flask. The cells were incubated at 37°C in a 5% $CO_2$ atmosphere for 10 minutes.

Hypotonic stimulation was performed by placing the flask on an orbital shaker inside the $CO_2$ incubator and adjusting the speed to 40 rpm. The stimulation was progressively introduced by adding 1.5, 1.8, or 2.2 ml of deionized water at 3-minute intervals. Following this, the speed of the orbital shaker was increased to 60 rpm for 5 minutes.

### eMigrasome purification

First, the bubbles were harvested using the following procedure. The supernatant containing cell bodies was discarded. The flask's bottom was gently washed once with hK-DPBS (60% K-DPBS matching the current osmolarity). Subsequently, 4 ml of hK-DPBS-BSA (60% K-DPBS containing 1 mg/ml BSA) was added to the flask. Gentle pipetting was employed to detach the eMigrasomes, and the resultant solution was collected into a conical tube. This step was repeated once, and the collected solutions were combined.

To remove the remaining cell bodies, a process of differential centrifugation and gravity-dependent filtration was used. Initially, the solution was centrifuged at 300 × *g* at 4°C for 10 minutes, after which the supernatant was collected. The same process was repeated at 500 × *g* for 10 minutes. The collected supernatant was poured into a filter cup containing a pre-rinsed 6 µm parylene filter (Hangzhou Branemagic Medical Technology, F-PAC007). The filter was pre-rinsed with 100% ethanol

followed by hK-DPBS-BSA. The flow-through was then collected in a low-protein-binding conical tube (Eppendorf, 0030122240).

To concentrate the eMigrasomes, the solution was centrifuged at 20,000 × $g$ at 4°C for 30 minutes. In the event of a large volume, a longer time may be necessary. The supernatant was discarded, and the eMigrasome pellet was resuspended in PBS. The protein concentration was determined using a BCA analysis, and the PBS volume was adjusted to reach the desired eMigrasome concentration. For long-term storage, it was recommended that the eMigrasomes were stored as a pellet, with a small volume of hK-DPBS-BSA to cover it. This measure was adopted to minimize eMigrasome loss due to container adsorption.

## Negative staining TEM

Isolated eMigrasomes underwent fixation by the addition of an equal volume of 2.5% glutaraldehyde (GA). A droplet of the fixed eMigrasome sample was deposited onto a copper grid for 15 minutes. Excess sample was blotted off with filter paper. The grid was promptly washed with a droplet of double distilled water (ddH$_2$O), stained with 1% uranyl acid for 1 minute, and then further washed with two droplets of ddH$_2$O. Remaining water was blotted off using filter paper and the grid was allowed to air-dry fully before imaging was conducted via transmission electron microscopy (TEM).

## Cryo-EM sample preparation and image acquisition

Quantifoil Cu grids (200 mesh, R2/2) underwent glow-discharging using a plasma cleaning device (PDC-32G, Harrick Plasma). Each EM grid was applied with 4 µl sample and vitrified by plugging into liquid ethane using the Vitrobot Mark IV system (Thermo Fisher Scientific). The cryo-EM samples were examined using an FEI Tecnai Arctica 200 kV transmission electron microscope, and images were captured at a magnification of 23.5 kx using an FEI Falcon II direct electron detector, with a dose approximately around 15 e−Å−2.

## eMigrasome quantification using flow cytometry

Isolated eMigrasomes were suspended in h-KDPBS containing 1 µg/ml Wheat Germ Agglutinin (WGA, Thermo, W11262). Various dilutions were prepared to ensure at least one dilution had a concentration ranging from 1000–10,000 eMigrasomes per µl. Data were collected using a CytoFlex LX cytometer (Beckman). The threshold for forward scatter (FSC) was manually set to 4000 to detect small particles. Events that were double positive for B525-FITC and Y610-mCherry were gated as eMigrasomes.

## Size analysis by Hough circle transforming

Size analysis of isolated eMigrasomes was conducted using ImageJ, complemented by the Hough circle transform plugin. Sum slices processing of Z-stack was applied to the 488 channel of the confocal images, representing the fluorescence signal from Tspan4-GFP. The image was subsequently converted to an 8-bit grayscale. Edge detection was performed using the 'Find Edges' function to identify individual eMigrasomes. Subsequently, the 'Threshold' function was applied to encompass the majority of the fluorescent signal from the eMigrasomes. Hough circle transform analysis was then utilized with the following parameters: easy mode; minimum = 3 pixels; maximum = 24 pixels; Hough score threshold = 0.9. All output options were selected.

## Immunoblotting

Cells or eMigrasomes were subjected to lysis using a 2% SDS solution in 50 mM Tris buffer, followed by heating at 95°C. The protein concentration of the samples was assessed using a BCA kit (Vazyme, E112-02-AB). The lysates were diluted to the desired concentration and then denatured via the addition of loading buffer (Beyotime, P0015). The proteins were segregated by SDS-PAGE electrophoresis (Epizyme, PG112, or Yeasen, 36255ES10), and subsequently transferred to a 0.45 µm PVDF membrane (Millipore, IPVH00010) in accordance with a standard protocol. The blot was blocked with 5% milk in TBST and left to incubate overnight with the primary antibody at 4°C. The blot was then washed thrice with TBST and incubated with the secondary antibody at RT for 1 hour. Finally, the blot underwent three more washes prior to signal detection via chemiluminescence imaging (CYANAGEN, XLS070P, or Thermo, 34075), utilizing a ChemiDoc MP Imaging System (Bio-Rad).

Primary antibodies used for immunoblotting included anti-integrin α5 (CST, 4705T), anti-Na-K-ATPase (CST, 3010S), anti-histone H3 (CST, 4499S), anti-lamp2 (Sigma), anti-calnexin (abcam, ab22595), anti-GFP (Roche, 11814460001 or abcam, ab290), anti-GAPDH (Proteintech, 60004-1-Ig), anti-TIM23 (BD, 611222), anti-S1 (Sinobio, 40591-MM42), anti-OVA (Santa Cruz, sc80587 or abcam, ab181688), and anti-mCherry (abcam, ab125096). Primary antibodies were diluted using Solution 1 (Takara, NKB-101). Secondary antibodies used for immunoblotting included peroxidase AffiniPure goat anti-rabbit IgG (H+L) (Jackson ImmunoResearch, 111-035-003) and peroxidase AffiniPure goat anti-mouse IgG (H+L) (Jackson ImmunoResearch,115-035-003). Secondary antibodies were diluted using 5% milk in TBST.

## Mice
WT C57BL/6 (Jax 000664) specific-pathogen-free (SPF) mice were procured from the Laboratory Animal Center of Tsinghua University, China. The OT-II (Jax 004194) mice were donated by Dr. Yan Shi. All the mice were bred and maintained under SPF conditions at the Laboratory Animal Center of Tsinghua University, in accordance with the National Institutes of Health Guide for the Care and Use of Laboratory Animals. The protocol was approved by the Institutional Animal Care and Use Committee (IACUC) of Tsinghua University (protocol #22-LZH1). For sample size determination, the number of mice was rationally determined based on the known effect size.

## Immunization
For systemic immunization, mice were intravenously administered eM-OVA dissolved in 100 µl PBS, or intraperitoneally administered a mixture containing 50 µg OVA in 50 µL PBS (Sigma; A5503) and 50 µl Imject Alum (Thermo; 77161). For nasal immunization, mice were anesthetized using isoflurane and subsequently intranasally administered 20 µg eM-OVA dissolved in 30 µl PBS. For subcutaneous immunization, 20 µg eM-OVA in 100 µl PBS was administered through injections on both sides of the buttock.

## Enzyme-linked immunosorbent assay (ELISA)
Serum was prepared from whole blood by centrifugation. The levels of antigen-specific antibody were determined using a direct ELISA method. Briefly, a 96-well plate was coated overnight at 4°C with antigen (2 µg/ml). The wells were then blocked with PBS containing 10% fetal calf serum before the addition of serially diluted serum. Horseradish peroxidase-conjugated secondary antibodies were incubated for 1 hour at RT. Between each step, wells were washed with PBST. The colorimetric reaction was carried out using the 1-Step Ultra TMB-ELISA Substrate. The reaction was stopped with 2 N $H_2SO_4$, and absorbance at 450 nm was read using a multimode reader. For quantification of OVA-IgG and Spike-IgG, a standard curve was generated using serially diluted anti-OVA antibody (Santa Cruz; sc-80589) and anti-Spike antibody (Sino Biological; 40591-MM42), respectively.

## Fluorescence-based neutralization test
Mouse sera were heat-inactivated at 56°C for a duration of 30 minutes. The indicated dilutions of samples were mixed with $2 \times 10^2$ FFU (focus-forming units) of VSV-Venus-SARS-CoV-2 and incubated for 1 hour at a temperature of 37°C. The mixture of serum and virus was then added to Vero-TMPRSS2 cells grown on 96-well plates and incubated at 37°C for about 40 hours. The cells were then harvested and fixed in a 4% paraformaldehyde solution for 20 minutes at RT. The fixed cells were resuspended in PBS and analyzed using a LSRFortessa SORP (BD Biosciences) and FlowJo software.

The Vero-TMPRSS2 cell line, which was constructed based on Vero (ATCC CCL-81), stably expressed the TMPRSS2 (Transmembrane Serine Protease 2) protein to enhance the entry of the SARS-CoV-2 spike.

## Statistics
All data were subjected to analysis using GraphPad Prism statistical software. Unpaired two-tailed *t*-tests or paired two-tailed *t*-tests were employed for the data analysis. The results are represented as the mean ± SD. A *P* value of <0.05 was deemed to indicate statistical significance. At least two biological replicates were performed for all the experiments in this study.

## Acknowledgements

This research was supported by the Ministry of Science and Technology of the People's Republic of China (grant no. 2024YFA1307301 to LY), Tsinghua-Toyota Joint Research Fund (grant no. 20233930058 to LY), Tsinghua University Dushi Program (grant no. 20251080019 to LY), the National Natural Science Foundation of China (grant nos. 32030023, 92354306, and 32330025 to LY, 82530021 to ZL), and Scientific and Technological Innovation Project of China Academy of Chinese Medical Sciences (grant no.CI2023C024YL to LY). Sincere gratitude is expressed to Baidong Hou (Institute of Biophysics, Chinese Academy of Sciences) for providing mouse strains and engaging in valuable discussions. We thank the State Key Laboratory of Membrane Biology, SLSTU-Nikon Biological Imaging Center, Laboratory of Animal Resources Center (THU-LARC), Tsinghua University, for technical support. We also acknowledge the Research Fund of the Vanke School of Public Health at Tsinghua University.

## Additional information

### Competing interests

Dongju Wang, Takami Sho, Yi Zheng: is one of the inventors on relevant patent applications (Patent No.: US 12221631 B2) held by Migrasome Therapeutics. Longyu Dou: employee of Migrasome Therapeutics. Li Yu: is the scientific founder of Migrasome Therapeutics. The other authors declare that no competing interests exist.

### Funding

| Funder | Grant reference number | Author |
| --- | --- | --- |
| Ministry of Science and Technology of the People's Republic of China | 2024YFA1307301 | Li Yu |
| Tsinghua-Toyota Joint Research Fund | 20233930058 | Li Yu |
| Tsinghua University | 20251080019 | Li Yu |
| National Natural Science Foundation of China | 32030023 | Li Yu |
| National Natural Science Foundation of China | 82530021 | Zhihua Liu |
| Scientific and Technological Innovation Project of China Academy of Chinese Medical Sciences | CI2023C024YL | Li Yu |
| National Natural Science Foundation of China | 92354306 | Li Yu |
| National Natural Science Foundation of China | 32330025 | Li Yu |

The funders had no role in study design, data collection and interpretation, or the decision to submit the work for publication.

### Author contributions

Dongju Wang, Haifang Wang, Conceptualization, Investigation, Methodology, Writing – original draft, Writing – review and editing; Wei Wan, Zihui Zhu, Investigation, Methodology; Takami Sho, Yi Zheng, Xing Zhang, Longyu Dou, Investigation; Qiang Ding, Supervision, Methodology; Li Yu, Zhihua Liu, Conceptualization, Supervision, Funding acquisition, Writing – original draft, Writing – review and editing

### Author ORCIDs

Dongju Wang https://orcid.org/0009-0007-1409-8482

Li Yu https://orcid.org/0000-0002-3757-0758
Zhihua Liu https://orcid.org/0000-0002-0269-0901

### Ethics

All the mice were bred and maintained under SPF conditions at the Laboratory Animal Center of Tsinghua University, in accordance with the National Institute of Health Guide for the Care and Use of Laboratory Animals. The protocol was approved by the Institutional Animal Care and Use Committee (IACUC) of Tsinghua University (Protocol #22-LZH1).

Reviewer #1 (Public review): https://doi.org/10.7554/eLife.97621.3.sa1
Reviewer #2 (Public review): https://doi.org/10.7554/eLife.97621.3.sa2
Author response https://doi.org/10.7554/eLife.97621.3.sa3

---

## Additional files

### Supplementary files

MDAR checklist

### Data availability

All data generated or analysed during this study are included in the manuscript, figures, figure supplements, and source data files.

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
