## [Editor Report · eLife Assessment]

This study, from the group that pioneered migrasome, describes a novel vaccine platform of engineered migrosomes that behave like natural migrasomes. Importantly, this platform has the potential to overcome obstacles associated with cold chain issues for vaccines such as mRNA. In the revised version, the authors have addressed previous concerns, and the results from additional experiments provide **compelling** evidence that features methods, data, and analyses more rigorous than the current state-of-the-art. Although the findings are **important** with practical implications for the vaccine technology, results from additional experiments would make this an outstanding study.

---

## [Referee Report · Reviewer #1 (Public review)]

Summary:

Outstanding fundamental phenomenon (migrasomes) en route to become transitionally highly significant.

Strengths:

Innovative approach at several levels: Migrasomes, discovered by DR. Yu's group, are an outstanding biological phenomenon of fundamental interest and now of potentially practical value.

Weaknesses:

I feel that the overemphasis on practical aspects (vaccine), however important, eclipses some of the fundamental aspects that may be just as important and actually more interesting. If this can be expanded, the study would be outstanding.

Comments on revisions: This reviewer feels that the authors have addressed all issues.

---

## [Referee Report · Reviewer #2 (Public review)]

Summary:

The authors report describes a novel vaccine platform derived from a newly discovered organelle called a migrasome. First, the authors address a technical hurdle for using migrasomes as a vaccine platform. Natural migrasome formation occurs at low levels and is labor intensive, however, by understanding the molecular underpinning of migrasome formation, the authors have designed a method to make engineered migrasomes from cultures cells at higher yields utilizing a robust process. These engineered migrasomes behave like natural migrasomes. Next, the authors immunized mice with migrasomes that either expressed a model peptide or the SARS-CoV-2 spike protein. Antibodies against the spike protein were raised that could be boosted by a 2nd vaccination and these antibodies were functional as assessed by an in vitro pseudoviral assay. This new vaccine platform has the potential to overcome obstacles such as cold chain issues for vaccines like messenger RNA that require very stringent storage conditions.

Strengths:

The authors present very robust studies detailing the biology behind migrasome formation and this fundamental understanding was used to from engineered migrasomes, which makes it possible to utilize migrasomes as a vaccine platform. The characterization of engineered migrasomes is thorough and establishes comparability with naturally occurring migrasomes. The biophysical characterization of the migrasomes is well done, including thermal stability and characterization of the particle size (important characterizations for a good vaccine).

Weaknesses:

With a new vaccine platform technology, it would be nice to compare them head-to-head against a proven technology. The authors would improve the manuscript if they made some comparisons to other vaccine platforms such as a SARS-CoV-2 mRNA vaccine or even an adjuvanted recombinant spike protein. This would demonstrate a migrasome based vaccine could elicit responses comparable to a proven vaccine technology. Additionally, understanding the integrity of the antigens expressed in their migrasomes could be useful. This could be done by looking at functional monoclonal antibody binding to their migrasomes in a confocal microscopy experiment.

Updates after revision:

The revised manuscript has additional experiments that I believe improve the strength of evidence presented in the manuscript and address the weaknesses of the first draft. First, they provide a comparison to the antibody responses induced by their migrasome based platform to recombinant protein formulated in an adjuvant and show the response is comparable. Second, they provide evidence that the spike protein incorporated into their migrasomes retains structural integrity by preserving binding to monoclonal antibodies. Together, these results strengthen the paper significantly and support the claims that the novel migrasome based vaccine platform could be a useful in the vaccine development field.

---

## [Author Response]

The following is the authors’ response to the original reviews

**Public Reviews:**

**Reviewer #1 (Public Review):**
Summary:This is an excellent study by a superb investigator who discovered and is championing the field of migrasomes. This study contains a hidden "gem" - the induction of migrasomes by hypotonicity and how that happens. In summary, an outstanding fundamental phenomenon (migrasomes) en route to becoming transitionally highly significant.Strengths:Innovative approach at several levels. Migrasomes - discovered by Dr Yu's group - are an outstanding biological phenomenon of fundamental interest and now of potentially practical value.Weaknesses:I feel that the overemphasis on practical aspects (vaccine), however important, eclipses some of the fundamental aspects that may be just as important and actually more interesting. If this can be expanded, the study would be outstanding.

We sincerely thank the reviewer for the encouraging and insightful comments. We fully agree that the fundamental aspects of migrasome biology are of great importance and deserve deeper exploration.

In line with the reviewer’s suggestion, we have expanded our discussion on the basic biology of engineered migrasomes (eMigs). A recent study by the Okochi group at the Tokyo Institute of Technology demonstrated that hypoosmotic stress induces the formation of migrasome-like vesicles, involving cytoplasmic influx and requiring cholesterol for their formation (DOI: 10.1002/1873-3468.14816, February 2024). Building on this, our study provides a detailed characterization of hypoosmotic stressinduced eMig formation, and further compares the biophysical properties of natural migrasomes and eMigs. Notably, the inherent stability of eMigs makes them particularly promising as a vaccine platform.

Finally, we would like to note that our laboratory continues to investigate multiple aspects of migrasome biology. In collaboration with our colleagues, we recently completed a study elucidating the mechanical forces involved in migrasome formation (DOI: 10.1016/j.bpj.2024.12.029), which further complements the findings presented here.

**Reviewer #2 (Public review):**
Summary:The authors' report describes a novel vaccine platform derived from a newly discovered organelle called a migrasome. First, the authors address a technical hurdle in using migrasomes as a vaccine platform. Natural migrasome formation occurs at low levels and is labor intensive, however, by understanding the molecular underpinning of migrasome formation, the authors have designed a method to make engineered migrasomes from cultured, cells at higher yields utilizing a robust process. These engineered migrasomes behave like natural migrasomes. Next, the authors immunized mice with migrasomes that either expressed a model peptide or the SARSCoV-2 spike protein. Antibodies against the spike protein were raised that could be boosted by a 2nd vaccination and these antibodies were functional as assessed by an in vitro pseudoviral assay. This new vaccine platform has the potential to overcome obstacles such as cold chain issues for vaccines like messenger RNA that require very stringent storage conditions.Strengths:The authors present very robust studies detailing the biology behind migrasome formation and this fundamental understanding was used to form engineered migrasomes, which makes it possible to utilize migrasomes as a vaccine platform. The characterization of engineered migrasomes is thorough and establishes comparability with naturally occurring migrasomes. The biophysical characterization of the migrasomes is well done including thermal stability and characterization of the particle size (important characterizations for a good vaccine).Weaknesses:With a new vaccine platform technology, it would be nice to compare them head-tohead against a proven technology. The authors would improve the manuscript if they made some comparisons to other vaccine platforms such as a SARS-CoV-2 mRNA vaccine or even an adjuvanted recombinant spike protein. This would demonstrate a migrasome-based vaccine could elicit responses comparable to a proven vaccine technology.

We thank the reviewer for the thoughtful evaluation and constructive suggestions, which have helped us strengthen the manuscript.

Comparison with proven vaccine technologies:

In response to the reviewer’s comment, we now include a direct comparison of the antibody responses elicited by eMig-Spike and a conventional recombinant S1 protein vaccine formulated with Alum. As shown in the revised manuscript (Author response image 1), the levels of S1-specific IgG induced by the eMig-based platform were comparable to those induced by the S1+Alum formulation. This comparison supports the potential of eMigs as a competitive alternative to established vaccine platforms.

**Author response image 1. sa3fig1:** eMigrasome-based vaccination showed similar efficacy compared with adjuvanted recombinant spike protein The amount of S1-specific IgG in mouse serum was quantified by ELISA on day 14 after immunization. Mice were either intraperitoneally (i.p.) immunized with recombinant Alum/S1 or intravenously (i.v.) immunized with eM-NC, eM-S or recombinant S1. The administered doses were 20 µg/mouse for eMigrasomes, 10 µg/mouse (i.v.) or 50 µg/mouse (i.p.) for recombinant S1 and 50 µl/mouse for Aluminium adjuvant.

Assessment of antigen integrity on migrasomes:

To address the reviewer’s suggestion regarding antigen integrity, we performed immunoblotting using antibodies against both S1 and mCherry. Two distinct bands were observed: one at the expected molecular weight of the S-mCherry fusion protein, and a higher molecular weight band that may represent oligomerized or higher-order forms of the Spike protein (Figure 5B in the revised manuscript).

Furthermore, we performed confocal microscopy using a monoclonal antibody against Spike (anti-S). Co-localization analysis revealed strong overlap between the mCherry fluorescence and anti-Spike staining, confirming the proper presentation and surface localization of intact S-mCherry fusion protein on eMigrasomes (Figure 5C in the revised manuscript). These results confirm the structural integrity and antigenic fidelity of the Spike protein expressed on eMigrasomes.

**Recommendations for the authors**

**Reviewer #1 (Recommendations For The Authors):**
I feel that the overemphasis on practical aspects (vaccine), however important, eclipses some of the fundamental aspects that may be just as important and actually more interesting. If this can be expanded, the study would be outstanding.I know that the reviewers always ask for more, and this is not the case here. Can the abstract and title be changed to emphasize the science behind migrasome formation, and possibly add a few more fundamental aspects on how hypotonic shock induces migrasomes?Alternatively, if the authors desire to maintain the emphasis on vaccines, can immunological mechanisms be somewhat expanded in order to - at least to some extent - explain why migrasomes are a better vaccine vehicle?One way or another, this reviewer is highly supportive of this study and it is really up to the authors and the editor to decide whether my comments are of use or not.My recommendation is to go ahead with publishing after some adjustments as per above.

We’d like to thank the reviewer for the suggestion. We have changed the title of the manuscript and modified the abstract, emphasizing the fundamental science behind the development of eMigrasome. To gain some immunological information on eMig illucidated antibody responses, we characterized the type of IgG induced by eM-OVA in mice, and compared it to that induced by Alum/OVA. The IgG response to Alum/OVA was dominated by IgG1. Quite differently, eM-OVA induced an even distribution of IgG subtypes, including IgG1, IgG2b, IgG2c, and IgG3 (Figure 4I in the revised manuscript). The ratio between IgG1 and IgG2a/c indicates a Th1 or Th2 type humoral immune response. Thus, eM-OVA immunization induces a balance of Th1/Th2 immune responses.

**Reviewer #2 (Recommendations For The Authors):**
The study is a very nice exploration of a new vaccine platform. This reviewer believes that a more head-to-head comparison to the current vaccine SARS-CoV-2 vaccine platform would improve the manuscript. This comparison is done with OVA antigen, but this model antigen is not as exciting as a functional head-to-head with a SARS-CoV-2 vaccine.

I think that two other discussion points should be included in the manuscript. First, was the host-cell protein evaluated? If not, I would include that point on how issues of host cell contamination of the migrasome could play a role in the responses and safety of a vaccine. Second, I would discuss antigen incorporation and localization into the platform. For example, the full-length spike being expressed has a native signal peptide and transmembrane domain. The authors point out that a transmembrane domain can be added to display an antigen that does not have one natively expressed, however, without a signal peptide this would not be secreted and localized properly. I would suggest adding a discussion of how a non-native signal peptide would be necessary in addition to a transmembrane domain.

We thank the reviewer for these thoughtful suggestions and fully agree that the points raised are important for the translational development of eMig-based vaccines.

(1) Host cell proteins and potential immunogenicity:

We appreciate the reviewer’s suggestion to consider host cell protein contamination. Considering potential clinical application of eMigrasomes in the future, we will use human cells with low immunogenicity such as HEK-293 or embryonic stem cells (ESCs) to generate eMigrasomes. Also, we will follow a QC that meets the standard of validated EV-based vaccination techniques.

(2) Antigen incorporation and localization—signal peptide and transmembrane domain:

We also agree with the reviewer’s point that proper surface display of antigens on eMigs requires both a transmembrane domain and a signal peptide for correct trafficking and membrane anchoring. For instance, in the case of full-length Spike protein, the native signal peptide and transmembrane domain ensure proper localization to the plasma membrane and subsequent incorporation into eMigs. In case of OVA, a secretary protein that contains a native signal peptide yet lacks a transmembrane domain, an engineered transmembrane domain is required. For antigens that do not naturally contain these features, both a non-native signal peptide and an artificial transmembrane domain are necessary. We have clarified this point in the revised discussion and explicitly noted the requirement for a signal peptide when engineering antigens for surface display on migrasomes.